# Influence of InP/ZnS Quantum Dots on Thermodynamic Properties and Morphology of the DPPC/DPPG Monolayers at Different Temperatures

**DOI:** 10.3390/molecules28031118

**Published:** 2023-01-22

**Authors:** Juan Wang, Shun Feng, Qingqing Sheng, Ruilin Liu

**Affiliations:** 1Shaanxi Engineering Research Center of Controllable Neutron Source, School of Electronic Information, Xijing University, Xi’an 710123, China; 2School of Pharmacy, Xuzhou Medical University, Xuzhou 221004, China

**Keywords:** InP/ZnS quantum dots, Langmuir monolayer, DPPC/DPPG mixed monolayer, atomic force microscope

## Abstract

In this work, the effects of InP/ZnS quantum dots modified with amino or carboxyl group on the characteristic parameters in phase behavior, elastic modulus, relaxation time of the DPPC/DPPG mixed monolayers are studied by the Langmuir technology at the temperature of 37, 40 and 45 °C. Additionally, the information on the morphology and height of monolayers are obtained by the Langmuir–Bloggett technique and atomic force microscope technique. The results suggest that the modification of the groups can reduce the compressibility of monolayers at a higher temperature, and the most significant effect is the role of the amino group. At a high temperature of 45 °C, the penetration ability of InP/ZnS-NH_2_ quantum dots in the LC phase of the mixed monolayer is stronger. At 37 °C and 40 °C, there is no clear difference between the penetration ability of InP/ZnS-NH_2_ quantum dots and InP/ZnS-COOH quantum dots. The InP/ZnS-NH_2_ quantum dots can prolong the recombination of monolayers at 45 °C and accelerate it at 37 °C and 40 °C either in the LE phase or in the LC phase. However, the InP/ZnS-COOH quantum dots can accelerate it in the LE phase at all temperatures involved but only prolong it at 45 °C in the LC phase. This work provides support for understanding the effects of InP/ZnS nanoparticles on the structure and properties of cell membranes, which is useful for understanding the behavior about the ingestion of nanoparticles by cells and the cause of toxicity.

## 1. Introduction

Quantum dots (QDs) are a very interesting class of nanomaterials with important applications in medical imaging and therapy. The most popular and currently well studied quantum dots are CdSe and CdTe, but they are highly toxic and carcinogenic to living systems. In order to reduce their toxicity, ZnS is usually used as shells to wrap cadmium-containing quantum dots to form core–shell quantum dots, such as CdSe/ZnS QDs. However, the CdSe/ZnS QDs are still toxic, which is related to the release of toxic Cd^2+^ ions [1]. Therefore, whether cadmium-containing quantum dots will be approved for medical use remains a highly controversial issue. Indium phosphide (InP) quantum dots are expected to be one of the best substitutes for cadmium-based quantum dots due to their advantages of heavy-metal-free and adjustable luminescence [2,3]. However, InP quantum dots have surface defects, which are not conducive to the generation of multi-exciton [4,5]. Using ZnS to cover InP quantum dots to form core–shell quantum dots, the multi-exciton lifetime of InP quantum dots can be controlled and luminescence can be enhanced [6,7,8]. Therefore, InP quantum dots in core–shell form, such as InP/ZnS, are more likely to be applied in the biomedical field due to their advantages of high luminescence and low cytotoxicity. The cellular uptake and localization of the CdSe/ZnS QDs and InP/ZnS QDs are identical. Under the same conditions, the toxicity of InP QDs is significantly reduced compared with CdSe QDs [9]. Water-soluble InP/ZnS core/shell QDs are safer than CdSe/ZnS QDs in biological applications. Although InP/ZnS quantum dots can replace cadmium-based particles to reduce the risk compared with cadmium-containing quantum dots, the understanding of toxicity of InP quantum dots is still in its infancy. The toxicological effects of InP quantum dots are also rarely studied.

The inhalation of nanoparticles into the lungs has a potential threat to human health. The alveolar surface is the first exposure target of inhaled nanoparticles in the human body. Information on the interaction between nanoparticles and alveolar surface membranes is useful for understanding the behavior about the ingestion of nanoparticles by cells and the cause of toxicity. Additionally, the toxicity of nanoparticles depends on their physical and chemical properties and environmental conditions, such as surface modification [10]. Researchers [11] investigated the toxicity of InP/ZnS modified with different surface groups (COOH, NH2 and OH, respectively) in vitro by two common cell models of nanoparticle respiratory toxicity [12,13]: human lung cancer cell HCC-15 and alveolar epithelial-type (AEII) cell RLE-6TN. It was indicated that InP/ZnS-COOH and InP/ZnS-NH_2_ were easier to enter cells than InP/ZnS-OH. Whereas the above studies at the cellular level struggle to reflect the interaction mechanisms between InP/ZnS DQs and alveolar surface membranes, the monolayer model provides a method for understanding the interaction of quantum dots with cell membranes at the membrane level, which is at a smaller scale compared with the cellular level. Because of the structural complexity of cell membranes, simple models are needed to study the basic membrane processes. Langmuir monolayer [14,15] is a very famous model of biological membrane, which can precisely control the components and environment of monolayers. It is very suitable for studying the interaction between the cell membrane and nanoparticles [16].

Pulmonary surfactant coats the surface of the alveolar, and dipalmitoylphosphatidylcholine (DPPC) and dipalmitoylphosphatidylgly-cerol (DPPG) are the two main phospholipids of pulmonary surfactants. The DPPC molecules can reduce the surface tension, but they limit the fluidity of the monolayer, which causes a poor spreadability of pulmonary surfactants monolayer [17]. Luckily, the DPPG molecules can enhance the fluidity of the monolayer and make the monolayer rapidly spread [18]. The mixed monolayer composed of DPPC and DPPG (4:1, mol:mol) was widely adopted to mimic the real pulmonary surfactant monolayer [19,20] according to the compositional analysis of mammalian lung surfactant extracts.

In the previous study [21], using the Langmuir monolayer method, we found that the effects of InP/ZnS quantum dots coated by the -COOH or -NH_2_ groups on the thermodynamic characteristics and microstructure of pulmonary surfactant membrane were significantly different at 35 °C. However, the change in temperature can affect the physicochemical properties of cell membranes, such as the fluidity of membrane [22]. The main reason is that the conformation and distribution of phospholipid molecules in the membrane may be different at different temperatures [23]. Additionally, the effect of nanoparticles on the cell membranes is also different due to the change in temperature. For example, Fe_3_O_4_ NPs can make cell membranes denser at the low temperature, but the opposite is true at the high temperature [24]. When human tissues become inflamed or diseased, human body temperature is higher than normal life temperature. Additionally, when a tumor patient is treated for hyperthermia, the temperature of the tissue rises to 40–45 °C [25,26]. In particular, the current photothermal therapy of cancer requires a tissue temperature of 42–45 °C [27], which is a targeted non-invasive cancer treatment method. The InP/ZnS quantum dots may be used as a photosensitizer in the future and injected into the body to assist photothermal therapy. If the InP/ZnS quantum dots becomes toxic to cells, they will first interact with the cell membrane. The effect of quantum dots on the structure and properties of cell membranes may vary depending on the ambient temperature. When the human body is at normal physiological temperature or higher than normal physiological temperature, the study on the effect of quantum dots on the pulmonary surfactant monolayer is closer to the real situation in explaining the mechanism of the respiratory toxicity of fluorescent quantum dots to humans. Additionally, it is more beneficial to understand the cytotoxicity of the InP/ZnS quantum dots on the cells at the membrane level when they are used in biomedical imaging and therapy in the future.

Therefore, this paper studied the effect of InP/ZnS, InP/ZnS-NH_2_ and InP/ZnS-COOH nanoparticles on the surface pressure–mean molecular area isotherm and elastic modulus of DPPC/DPPG monolayer at 37 °C, 40 °C and 45 °C. Additionally, the relaxation and morphology of the mixed monolayer in the LE and LC phase in the presence of these nanoparticles has been researched. The results can help to understand the effect of InP/ZnS quantum dots coated by carboxyl group and amino group on cell membranes at different temperatures and to provide more abundant information for the future application of InP/ZnS quantum dots in vivo. 

## 2. Results and Discussion

### 2.1. Surface Pressure–Mean Molecular Area (π−A) Isotherms and Elastic Modulus–Surface Pressure ($C_{s}^{-1}-\pi$) Curves

Molecules in monolayers can exist in different states as a function of density. They are generally classified as gaseous, liquid-expanded (LE), liquid-condensed (LC), solid and intermediate or transition films, in many respects analogous to those in three-dimensional systems [28,29]. The higher the surface pressure, the denser the molecular arrangement, so the π−A isotherms show a continuous rise during compression. The π−A isotherms of the DPPC/DPPG (4:1, mol:mol) mixed monolayer at various temperatures on the surface of different subphase solutions are shown in Figure 1. The obvious plateaus that corresponded to the two-phase coexistence region are not observed on the π−A isotherm of the DPPC/DPPG mixed monolayer on the buffer without InP/ZnS QDs at 37 °C, which is similar to the literature [21,30,31,32]. When the InP/ZnS-NH_2_ and InP/ZnS-COOH QDs are present, respectively, in the subphase (Figure 1C,D), the shape of the π−A isotherms for the DPPC/DPPG mixed monolayer are similar to those in previous work [21]. The difference of the mean molecular area corresponding to the starting point of the rise of the π−A isotherm is mainly due to the different temperatures [33,34]. At higher surface pressure, the π−A isotherms show an obvious inflection point, which corresponded to a plateau in the isotherms, indicating that the monolayer began to collapse gradually. This is mainly because the extrusion of DPPG causes the monolayer to bend during compression, and the monolayers may break and accumulate to form multilayer films. The surface pressure at the inflection point is called collapse pressure ($\pi_{C}$), which is an important characteristic parameter of the π−A isotherm. The mean molecular area at collapse is marked as $A_{C}$. In addition, there are two characteristic parameters, respectively, the liftoff area $A_{L}$ (the molecular occupation area where the isotherm rise just emerges related to the baseline) and the limiting area $A_{\infty }$ (an empirical parameter approximating the mean molecular cross sectional area) [28,29]. The $A_{\infty }$ value is conventionally given from the π−A isotherms by extrapolating the slope of isotherm in its steepest range to the zero-surface pressure [35], if a liquid-condensed or solid phase is observed in the isotherm.

From Table 1, the $A_{L}$ value of the DPPC/DPPG mixed monolayer increases as the temperature rises in the absence of InP/ZnS QDs. This is mainly due to the fact that the movement rate of lipid molecules is faster and the arrangement of lipid molecules on the monolayer is irregular at higher temperatures, which leads to the increase of the occupied area of a molecule. In the presence of InP/ZnS-COOH QDs, the change in $A_{L}$ value with temperature is similar to that in the absence of quantum dots. However, in the presence of InP/ZnS-NH_2_ QDs, the $A_{L}$ value hardly changes with temperature. Additionally, in the presence of InP/ZnS QDs without modified groups, the $A_{L}$ value has no change when the temperature increases from 37 °C to 40 °C, but the $A_{L}$ value increases at 45 °C. These results indicate that the presence of the amino group makes InP/ZnS quantum dots significantly interfere with the change rule of the phospholipid molecular area with temperatures at the low surface pressure compared with carboxyl. On the contrary, the presence of the carboxyl group may avoid the influence of unmodified InP/ZnS QDs on the change rule of phospholipid molecular area when the temperature varies from 37 °C to 40 °C. However, the $A_{\infty }$ value and $A_{C}$ value all increase as the temperature rises and the $\pi_{C}$ value decreases as the temperature rises, which do not depend on the presence of InP/ZnS QDs and the modified groups on the surface of InP/ZnS QDs. This shows that when the temperature increases from 37 °C to 45 °C, the existence of quantum dots may not change the influence rule of temperature on these three values.

Direct determination of the surface phase state of the DPPC/DPPG mixed monolayer from the π−A isotherms is difficult because these isotherms have no sharp phase transitions. The phase of monolayers can be determined by the elastic modulus of the monolayer (Figure 1 insert), which can be calculated by the formula [36,37]:(1)$$C_{s}^{-1}=-A{\left(\frac{d\pi }{dA}\right)}_{T}$$
where $C_{s}^{-1}$ is the elastic modulus of the DPPC/DPPG mixed monolayer, *s* is the cross-sectional area of the monolayer, A is the mean molecular area and π is the surface pressure of the monolayer. A greater elastic modulus indicates that the monolayer is less compressible [38]. The minimum of $C_{s}^{-1}$ suggests a significant phase transition in the monolayer [39].

According to the early studies by J. T. Davies et al. [36], the value of $C_{s}^{-1}$ can classify the physical state of the lipid monolayer and the details are as follows [33,36,40,41]. When $C_{s}^{-1}=$0–12.5 mN/m, the monolayer is in gas (G) phase. When $C_{s}^{-1}=$12.5–50 mN/m, the monolayer is in liquid expansion (LE) phase. When $C_{s}^{-1}=$100–250 mN/m, the monolayer is in the liquid condensed (LC) phase. When $C_{s}^{-1}>250$ mN/m, the monolayer is in condensed (C) phase. From the Figure 1 insert, the phase state of the DPPC/DPPG mixed monolayer changes from G phase to LC phase with the increase of surface pressure. The minimum of $C_{s}^{-1}$ at low surface pressure means a phase transition from the G or LE phase to the LC phase of the monolayer, and the point at the minimum of $C_{s}^{-1}$ is called the phase transition point (point M). The curve of surface pressure corresponding to the point M changing with temperature is shown in Figure 2A. In the absence of InP/ZnS QDs, the surface pressure needed for the transition from LE phase to LC phase has little change in temperature, which is similar with that in the presence of InP/ZnS QDs without modified groups. Additionally, at the same temperature, InP/ZnS QDs do not influence the surface pressure needed for the transition. Differently, when the surface of InP/ZnS QDs is modified with the carboxyl group, the surface pressure needed for the transition increases at 40 °C. More interestingly, the amino group on the surface of InP/ZnS QDs makes the surface pressure needed for the transition significantly lower than that in other cases. This is due to the direct transition of the monolayer from G to LC phase in the presence of InP/ZnS-NH_2_ nanoparticles. Meanwhile, the higher the temperature, the lower the surface pressure needed for the transition from G phase to LC phase.

The maximum elastic modulus corresponds to the minimum of the compressibility during the compression of the monolayer (Figure 2B). In the absence of InP/ZnS QDs, the maximum of $C_{s}^{-1}$ increases gradually with the rise in temperature, which is similar to that in the presence of InP/ZnS-COOH QDs but is opposite to that in the presence of unmodified InP/ZnS QDs. At 37 °C, the InP/ZnS QDs modified by the -NH_2_ or -COOH group and the unmodified InP/ZnS QDs all make the maximum of $C_{s}^{-1}$ greater than that in the absence of InP/ZnS QDs, which is similar with that at 40 °C. Differently, the InP/ZnS QDs without a modified group decrease the maximum of $C_{s}^{-1}$, but the InP/ZnS QDs modified by -NH_2_ or -COOH group increase it at 45 °C, and the maximum of $C_{s}^{-1}$ is the most in the presence of the amino group. The modification of the groups can reduce the compressibility of the monolayer at a higher temperature, and the most significant effect is the role of the amino group.

### 2.2. Relaxation of the DPPC/DPPG Mixed Monolayers at Constant Area

To further understand the effect of InP/ZnS QDs on the stability of the DPPC/DPPG mixed monolayers, a relaxation study was performed at initial surface pressure ($\pi_{0}$) of 10 and 40 mN/m, where corresponded to the LE phase and LC phase of the monolayer. The surface pressure is recorded as a function of time at a constant area (π−t curve), which suggests the relaxation process of monolayers [28]. The π−t curves can be normalized to $\pi /\pi_{0}-t$ curves (seen in Figure 3), which can be fitted well by using the following equation [28,42]:(2)$$\pi /\pi_{0}=C+ae^{-t/\tau }$$
where *C* could be defined as the normalized equilibrium pressure and τ could be considered as the lifetime related to the reorganization of monolayers. The greater the value of τ, the longer the relaxation time of the monolayer, which suggests that the disorder degree of monolayers is higher, and the time of conformation transition is longer [28]. The parameter C and τ of the DPPC/DPPG mixed monolayer are listed in Table 2.

From the fitting results, the parameter C of the DPPC/DPPG mixed monolayer increases with the rise in temperature from 37 °C to 45 °C at the initial surface pressures of 10 mN/m and 40 mN/m, which does not depend on the presence of InP/ZnS QDs and the different modified groups of InP/ZnS QDs. It is consistent with the normalization of the equilibrium value of surface pressure ($\pi /\pi_{0}$) in experiments. The presence of InP/ZnS QDs influences the value of parameter τ. Compared with the pure buffer, the presence of InP/ZnS QDs without the modified group causes the value of τ decreased at the initial surface pressure of 10 mN/m at all the temperatures involved. At the initial surface pressure of 40 mN/m, it makes the value of τ decrease at 37 °C but increase at 40 °C and 45 °C. The presence of InP/ZnS QDs modified by the -NH_2_ group makes the value of τ decrease at 37 °C and 40 °C but increase at 45 °C when the initial surface pressure is 40 mN/m. This is similar with that in the presence of InP/ZnS QDs modified by the -COOH group. Differently, at the initial surface pressure of 10 mN/m, the presence of InP/ZnS-COOH QDs causes the value of τ to decrease at all the temperatures involved, but the presence of InP/ZnS-NH_2_ QDs makes it increase at 45 °C and decrease at 37 °C and 40 °C. The above is an analysis of the effect by different InP/ZnS quantum dots on the τ value compared with the situation in the absence of quantum dots. Compared with the unmodified InP/ZnS QDs, the influence of InP/ZnS QDs modified by different groups on the value of τ can be analyzed to understand the influence of different surface groups of quantum dots on the relaxation time of the DPPC/DPPG mixed monolayer. The influence of the -NH_2_ group and the -COOH group on the τ value are similar at the initial surface pressure of 40 mN/m, and they increase the τ value at 37 °C but decrease it at 40 °C and 45 °C compared with the unmodified InP/ZnS QDs. At the initial surface pressure of 10 mN/m, the -NH_2_ group decreases the τ value at 40 °C but increases it at 45 °C, which is similar with that of the -COOH group. Differently, at 37 °C, the -NH_2_ group decreases the τ value, while the -COOH group increases it.

### 2.3. The AFM Analysis of the Monolayer

According to the $C_{s}^{-1}$, the monolayer at 40 mN/m is in the LC phase, and the 10 mN/m monolayer is in the LE phase, which is confirmed by Figure 4. It can be observed that the DPPC/DPPG mixed monolayer is more compact at 40 mN/m than that at 10 mN/m, regardless of the presence of QDs in the subphase.

In the absence of QDs, the morphology of the DPPC/DPPG mixed monolayers is shown in Figure 4A. In the LE phase, the shape of the bright domains looks like “islands”, and the boundary of these domains is irregular at 37 °C, which is similar with the morphology of the DPPC/DPPG mixed monolayer in the environment of 35 °C [21]. As the temperature rises, the bright domains become more dispersed at 40 °C, and the bright areas appeared as some irregular and thick chains. At 45 °C, there are tiny and more dispersed regions around the “islands” areas. At the same time, the height analysis shows that the higher the temperature, the higher the height of the bright region. In the LC phase, the bright areas form a flat film with the black domains in the shape of “holes”. The higher the temperature, the larger the dark areas. The height of the mixed monolayer is about 2.58 nm at 37 °C, which closes to the theoretical value (2.8 nm) for the length of the DPPC molecules [43]. When the temperature rises, the height gradually decreases to about 2.2 nm at 45 °C, indicating an even smaller vertical orientation of the lipid molecules or the fat chain of phospholipids shrinks.

In the presence of the InP/ZnS nanoparticles without surface groups (Figure 4B), the morphology of the monolayer is significantly different from that in the absence of InP/ZnS nanoparticles in the LE phase. In Figure 4B(a–c), a spot-like bright region with a height of about 5–25 nm and a width of 30–70 nm is obviously observed, and both the height and width are integer multiples of the average diameter of the InP/ZnS nanoparticles, which proves that the InP/ZnS nanoparticles are inserted into the DPPC/DPPG mixed monolayer. It can be inferred that the InP/ZnS nanoparticles are more often clustered together and inserted into the monolayer in the LE phase (Figure 5). However, in the LC phase, no obvious spot-like bright region is observed, but the height of bright region reaches to 5–10 nm at 37 °C. When the temperature rises to 40 and 45 °C, the height of the bright region is about 5–6 nm. The morphology of the bright domain marked with a red line in Figure 4B(d) is similar to the bright domains in Figure 4B(e,f), which may be due to the adsorbing of InP/ZnS nanoparticles on the monolayer under the air–water interface. When the monolayer is transferred on the mica, the deposition of nanoparticles on the mica sheet results in their protrusion from the monolayer (Figure 5).

In the presence of InP/ZnS nanoparticles modified by the -NH_2_ group (Figure 4C) in the LE phase, some bright regions with heights of 5 nm are observed in the dark region at 37 and 40 °C, confirming the presence of InP/ZnS-NH_2_ nanoparticles. Compared with that at 37 °C, the width of the region with nanoparticles is smaller at 40 °C. However, at 45 °C, the nanoparticles appear in the bright region of the lipid monolayer with a height of about 10 nm and a width of about 300 nm. At the same time, it is observed that the morphology of the monolayer is gradually fragmented and more and more dispersed with the increase of temperature due to the existence of InP/ZnS-NH_2_ nanoparticles. In the LC phase, traces of nanoparticles with the height of 5–10 nm can be clearly observed on the mixed monolayer. The higher the temperature, the wider the region of monolayer adsorbed by the InP/ZnS-NH_2_ nanoparticles.

In the presence of the InP/ZnS nanoparticles modified by the -COOH group (Figure 4D) in the LE phase, the monolayer is homogeneous and dispersed. The height of the nanoparticles on the monolayer is about 5 nm at 37 °C but is 5–10 nm at 40 °C and 45 °C. In the LC phase, the height of the nanoparticles on the monolayer is about 5–10 nm at 37 °C and 40 °C but is 5 nm at 45 °C. However, the higher the temperature, the wider the region of monolayer adsorbed by the InP/ZnS-COOH nanoparticles, which is similar with the case in the presence of InP/ZnS-NH_2_ nanoparticles.

According to the analysis of AFM images, the diagram of the interaction between three types of InP/ZnS nanoparticles and the DPPC/DPPG mixed monolayers is shown in Figure 5 and Table 3. The InP/ZnS nanoparticles penetrate the monolayer as a single particle or as multiple particles aggregated together in the LE phase of the monolayer. Additionally, in the LC phase, the InP/ZnS nanoparticles mainly adsorb on the monolayer under the air–water interface at higher temperatures. However, when the surface of InP/ZnS nanoparticles is modified with the amino or carboxyl group, the interaction between nanoparticles and the monolayer is mainly manifested as their direct penetration into the monolayer. Differently at 45 °C, a large number of the InP/ZnS-NH_2_ nanoparticles aggregate together (corresponding to the height of the bright region around 10 nm) and penetrate into the monolayer in the LE phase. However, the InP/ZnS-COOH nanoparticles penetrate the monolayer in the form of a small amount of aggregation or single particles. When in the LC phase, InP/ZnS-NH_2_ nanoparticles still enter the monolayer in the form of aggregation or single particles with high density, but the form of aggregation disappears when the -COOH group replaces the -NH_2_ group at 45 °C. It can be speculated that the InP/ZnS nanoparticles are more likely to penetrate the cell membrane after being modified with amino and carboxyl groups. However, at a high temperature of 45 °C, the penetration ability of the amino-modified nanoparticles is stronger.

## 3. Materials and Methods

### 3.1. Materials

The 1,2-dipalmitoyl-sn-glycero-3-phosphocholine (DPPC: purity ≥ 99%) and 1,2-dipalmitoyl-sn-glycero-3-phosphoglycerol sodium (DPPG: purity ≥ 99%) used in this paper were obtained from Avanti Polar Lipids (Alabaster, AL). The aqueous InP/ZnS, InP/ZnS-NH_2_ and InP/ZnS-COOH QDs were purchased from Xi’an Qiyue Biological Technology Co., Ltd. (Xi’an, China), and the size of which was about 5 nm. The absorption spectra, photoluminescence spectra and TEM images of InP/ZnS QDs can be seen in supplementary data (Appendix A). The successful functionalization of the two surface groups (-NH_2_ and -COOH) has been confirmed and the sizes of the InP/ZnS-NH_2_ and InP/ZnS-COOH QDs have been shown in our previous work [21]. High purity water was produced by a Milli-Q plus water purification system (18.2 MΩ/cm, Millipore, Burlington, MA, USA). The 20 mM HEPES (N-2-hydroxyethylpiperazine-N-2-ethanesulfonic acid, pH 7.0) buffer was used as the subphase solution under the air–water interface in order to maintain the stability of pH value in the environment of the lipid monolayer.

### 3.2. Monolayers Preparation 

A Langmuir trough (KSV-Minitrough, Finland) with an area of 243 cm^2^ provides the air–water interface. A filter paper (10 mm × 30 mm × 0.15 mm), Wilhelmy-type tensiometer, was used to detect the surface behavior of the monolayer at the air–water interface, and the sensor accuracy was 0.01 mN/m. The DPPC and DPPG (4:1, mol:mol) were fully dissolved in chloroform/methanol (9:1, *v*/*v*) mixture, the final concentration of which was about 0.5 μmol/mL. We used a Hamilton microsyringe to spread the mixture on the air–water interface of subphase in the trough. We waited 20 min to make sure all organic solvents evaporate and then the Langmuir monolayers reach equilibrium. Then, the monolayers were used for further measurements. The experimental temperature was set to be 37 °C, 40 °C and 45 °C, respectively, which was maintained by a thermostatic circulating water bath apparatus with an accuracy of 0.05 °C in all experiments. 

### 3.3. Surface Pressure–Mean Molecular Area (π−A) Isotherms

Before spreading the lipid molecules on the interface, InP/ZnS, InP/ZnS-NH_2_ and InP/ZnS-COOH quantum dots were dispersed separately in the HEPES buffer solution of the trough with the same concentration of 0.2 μg/mL. In this way, the effect of InP/ZnS, InP/ZnS-NH_2_ and InP/ZnS-COOH quantum dots on the DPPC/DPPG mixed monolayer was studied. The DPPC/DPPG mixture was spread on the surface of the pure HEPES buffer or the solution with InP/ZnS, InP/ZnS-NH_2_ or InP/ZnS-COOH quantum dots. After 20 min, the monolayer was compressed with a constant rate of 7 mm/min, and surface pressure–mean molecular area (π−A) isotherms of the DPPC/DPPG mixed monolayer were obtained. Each measurement was repeated three times.

### 3.4. Relaxation of the DPPC/DPPG Mixed Monolayers at Constant Molecular Area

The DPPC/DPPG mixed monolayers at the air–water interface were compressed to the target surface pressure (5 mN/m, 20 mN/m and 40 mN/m) with a rate of 7 mm/min, and then the surface pressure–time π−t curves of the monolayers were recorded at a constant molecular area, which was relaxation progress in surface pressure vs. time. Each measurement was repeated three times.

### 3.5. Atomic Force Microscope Observation of the Monolayer 

The DPPC/DPPG mixed monolayers were deposited onto the fresh micas vertically at 10 mN/m and 40 mN/m at a constant dipping rate of 1 mm/min to form the Langmuir–Blodgett (LB) films by the Langmuir–Blodgett technology. Their morphology was observed by atomic force microscope (Shimadzu, Kyoto, Japan) in the intermittent contact mode using a silicon nitride pyramidal tip mounted on a 100 μm long cantilever with force constant of 0.1 N/m. 

## 4. Conclusions

When the DPPC/DPPG monolayer is compressed, the surface pressure required for the transition from LE to LC phase is almost not affected by the presence of InP/ZnS quantum dots at the same temperature, but it is affected by the InP/ZnS quantum dots with modified groups. Significantly differently, the InP/ZnS quantum dots with -NH_2_ group make the transition from LE to LC phase disappear but cause the transition from G to LC phase, and the higher the temperature, the lower the required surface pressure. According to the maximum of elastic modulus, the modification of groups can reduce the compressibility of monolayer at higher temperatures, and the most significant effect is in the presence of the amino group. The effect of nanoparticles on the relaxation of monolayer is affected by the surface groups, temperatures and phase state of the monolayer. From the AFM images, the InP/ZnS nanoparticles can penetrate the monolayer in the LE phase, which does not depend on the temperature. However, in the LC phase, the InP/ZnS nanoparticles mainly adsorb on the monolayer under the air–water interface at higher temperature. When the surface of InP/ZnS nanoparticles is modified with the amino or carboxyl group, the interaction between nanoparticles and the monolayer is mainly manifested as their direct penetration into the monolayer. Additionally, at a high temperature of 45 °C, the penetration ability of InP/ZnS-NH_2_ nanoparticles is stronger in the LC phase. Although the lipid monolayer model used in this study lacked the curvature of real cell membranes, the lipid monolayer model can help to precisely control the surface pressure and membrane environment, which was currently the most popular method for studying the interaction between nanoparticles, drugs and plasma membranes. When nanoparticles are used for in vivo imaging, the surface pressure in the local domains of the cell membrane changes as the nanoparticles are ingested by the cell membrane in vivo. In this process, changes in physiological temperature and different modification groups of nanoparticles may affect the interaction between nanoparticles and membranes. The results of this study provide detailed information on the effects of different InP/ZnS nanoparticles on the DPPC/DPPG mixed monolayers at different temperatures and surface pressures, which can provide support for understanding the effects of InP/ZnS nanoparticles on the structure and properties of cell membranes.

## Figures and Tables

**Figure 1 molecules-28-01118-f001:**
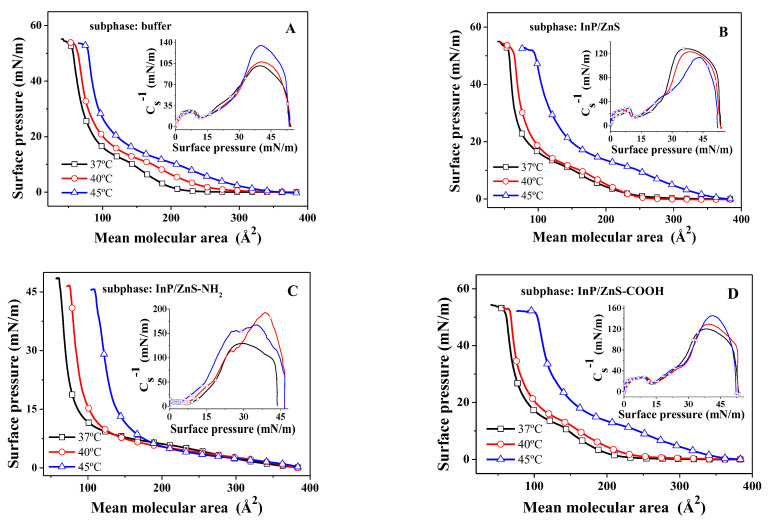
The surface pressure–mean molecular area (π−A) isotherms and the $C_{s}^{-1}-\pi$ curves (insert images) of the DPPC/DPPG mixed monolayer on the air–water interface in the absence of QDs (**A**) and in the presence of InP/ZnS (**B**), InP/ZnS-NH_2_ (**C**) and InP/ZnS-COOH QDs (**D**).

**Figure 2 molecules-28-01118-f002:**
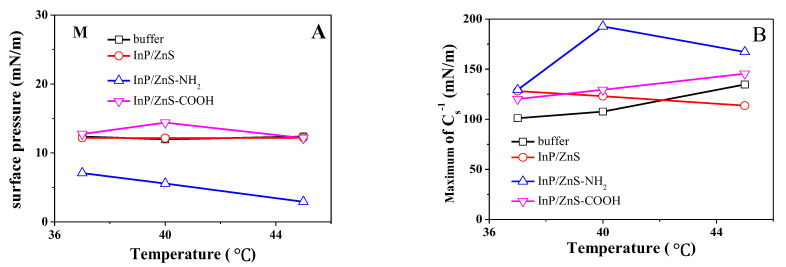
The surface pressure of phase transition point (**A**) and the maximum of $C_{s}^{-1}$ (**B**) for the DPPC/DPPG mixed monolayer at 37 °C, 40 °C and 45 °C.

**Figure 3 molecules-28-01118-f003:**
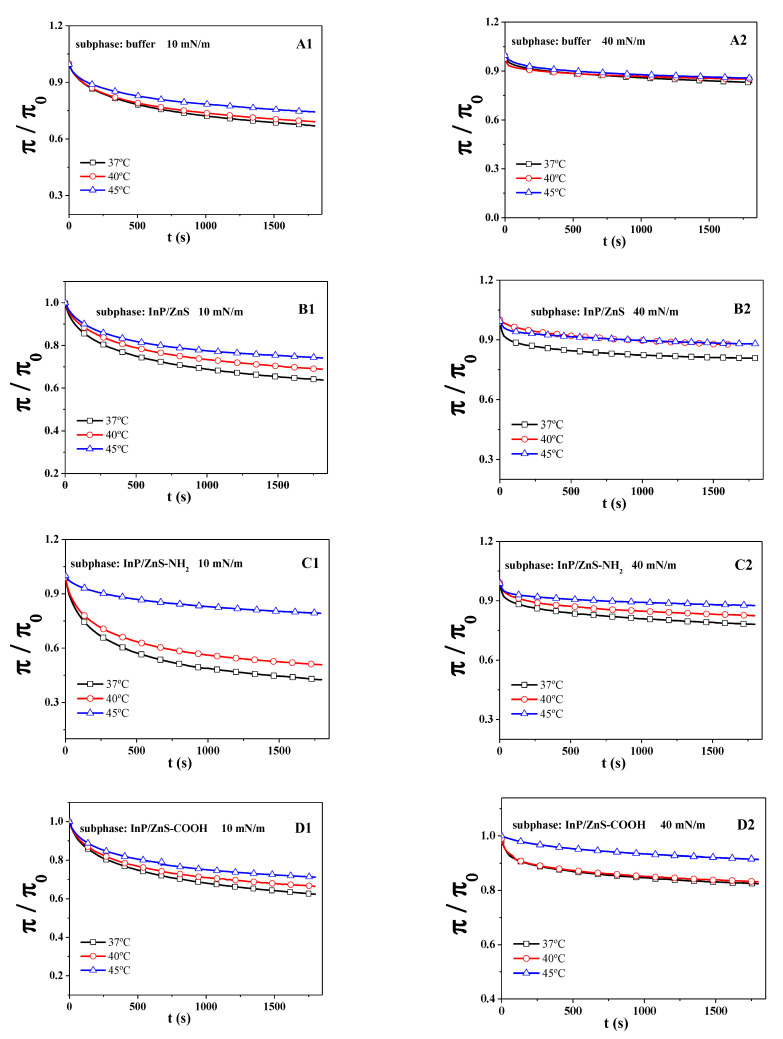
The $\pi /\pi_{0}-t$ curves of the DPPC/DPPG mixed monolayer in the absence of QDs (**A1**,**A2**) and in the presence of InP/ZnS (**B1**,**B2**), InP/ZnS-NH_2_ (**C1**,**C2**) and InP/ZnS-COOH QDs (**D1**,**D2**) at 37 °C, 40 °C and 45 °C.

**Figure 4 molecules-28-01118-f004:**
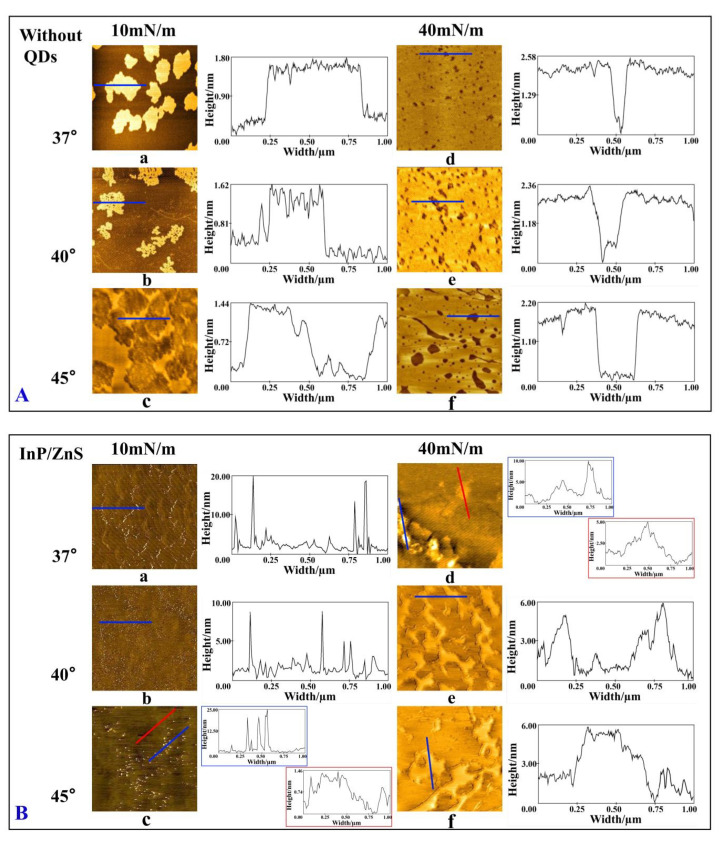
The morphology and height of the DPPC/DPPG mixed monolayer on the micas in the absence of QDs (**A**) and in the presence of InP/ZnS (**B**), InP/ZnS-NH_2_ (**C**) and InP/ZnS-COOH QDs (**D**) studied by atomic force microscope.

**Figure 5 molecules-28-01118-f005:**
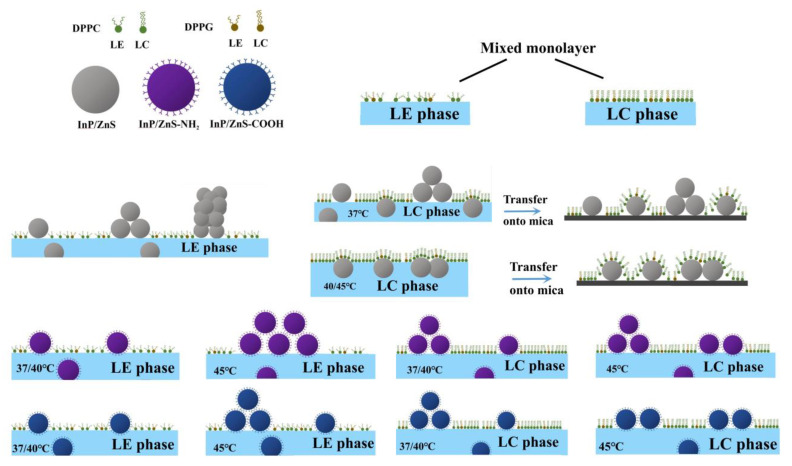
Diagram of interaction between three types of InP/ZnS nanoparticles and the DPPC/DPPG mixed monolayers.

**Table 1 molecules-28-01118-t001:** The three characteristic parameters and $A_{C}$ values at various temperatures on the surface of different subphase solutions ^a^.

Subphase	Temperature/°C	$A_{L}/{Å}^{2}$	$A_{\infty }/{Å}^{2}$	$\pi_{C}/mN/m$	$A_{C}/{Å}^{2}$
Without QDs	37	233.83	90.51	53.04	52.73
	40	293.53	97.32	52.76	60.87
	45	350.36	114.28	52.20	76.92
InP/ZnS	37	259.54	74.9	52.73	54.43
	40	259.54	84.16	51.79	62.37
	45	364.48	143.15	51.54	92.85
InP/ZnS-NH_2_	37	376.74	82.44	48.56	61.51
	40	376.74	102.72	46.95	74.65
	45	376.74	142.86	45.66	106.72
InP/ZnS-COOH	37	237.25	89.02	52.66	57.46
	40	271.98	94.22	52.63	65.76
	45	365.29	142.66	51.88	100.42

^a^ The errors in the values of $A_{L}$, $A_{\infty }\;$ and $A_{C}$ are all less than $\pm 0.50{Å}^{2}$. The errors in the value of $\pi_{C}$ are less than ±0.03 mN/m

**Table 2 molecules-28-01118-t002:** The values of $C,a,\tau ,\;r^{2}$, obtained by fitting the decay curves to a single-exponential equation.

Buffer Solution	T/°C	10 mN/m				40 mN/m			
		C	a	τ	$r^{2}$	C	a	τ	$r^{2}$
Without QDs	37	0.83	0.14	592.53	0.99	0.82	0.12	781.20	0.98
	40	0.85	0.13	553.45	0.99	0.85	0.08	591.03	0.98
	45	0.87	0.10	611.02	0.99	0.86	0.11	549.07	0.98
InP/ZnS	37	0.82	0.14	534.72	0.99	0.81	0.11	464.45	0.98
	40	0.84	0.13	546.15	0.99	0.87	0.11	705.51	0.99
	45	0.87	0.11	484.8	0.99	0.87	0.08	832.33	0.99
InP/ZnS-NH_2_	37	0.71	0.23	407.19	0.99	0.78	0.13	634.52	0.97
	40	0.76	0.18	448.64	0.99	0.82	0.11	505.87	0.97
	45	0.89	0.09	636.36	0.99	0.87	0.07	652.70	0.97
InP/ZnS-COOH	37	0.81	0.15	553.85	0.99	0.82	0.11	523.40	0.97
	40	0.83	0.14	523.05	0.99	0.83	0.11	530.77	0.96
	45	0.85	0.14	517.25	0.99	0.91	0.08	829.54	0.99

**Table 3 molecules-28-01118-t003:** The distribution of nanoparticles on the LE and LC phase of the DPPC/DPPG monolayer.

Buffer Solution	T/ °C	Distribution of Nanoparticles on the LE Phase of Monolayer		Distribution of Nanoparticles on the LC Phase of Monolayer	
		Position	Form	Position	Form
InP/ZnS	37	penetrate	Single, aggregated	Penetrate, adsorb on the interface	Single, aggregated
	40	penetrate	Single, aggregated	adsorb on the interface	Single, aggregated
	45	penetrate	Single, aggregated	adsorb on the interface	Single, aggregated
InP/ZnS-NH_2_	37	penetrate	Single	penetrate	Single, aggregated
	40	penetrate	Single	penetrate	Single, aggregated
	45	penetrate	aggregated	penetrate	aggregated
InP/ZnS-COOH	37	penetrate	Single	penetrate	Single, aggregated
	40	penetrate	Single	penetrate	aggregated
	45	penetrate	Single, aggregated	penetrate	aggregated

## Data Availability

Not applicable.

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
