# Peer review of "Influence of InP/ZnS Quantum Dots on Thermodynamic Properties and Morphology of the DPPC/DPPG Monolayers at Different Temperatures"

_molecules, 2023, doi:10.3390/molecules28031118_

Round 1

Reviewer 1 Report

The manuscript entitled “Influence of InP/ZnS Quantum Dots on Thermodynamic

Proper-Ties and Morphology of the DPPC/DPPG Monolayers at Dif-Ferent Temperatures”. Some issues to be addressed which will improve the quality of manuscript. Therefore, I recommend this work could be published after the major revision

1.      Should the author write down the novelty of this paper?

2.      The English composition requires many improvements. The authors should proofread the manuscript carefully to minimize grammatical errors.

3.      All the references mentioned in the paper should be cited in the text or vice-versa.

4.      The author, please add a comparative table for the reader's clear understanding of this work.

5.       Please increase the font size of the Y-axis in fig 3 so the reader can see them well.

6.      The conclusion look too much longer making its short up to a point.

Reviewer 2 Report

Find the attached file, please.

Round 2

Reviewer 1 Report

The author solve all comments very carefully, i recommended to accept in present form. 

Reviewer 2 Report

The revised version of the manuscript is improved well. The current form of the revised version may consider for acceptance.